# Comparison between Membrane and Thermal Dealcoholization Methods: Their Impact on the Chemical Parameters, Volatile Composition, and Sensory Characteristics of Wines

**DOI:** 10.3390/membranes11120957

**Published:** 2021-12-01

**Authors:** Faisal Eudes Sam, Tengzhen Ma, Yuhua Liang, Wenle Qiang, Richard Atinpoore Atuna, Francis Kweku Amagloh, Antonio Morata, Shunyu Han

**Affiliations:** 1Gansu Key Laboratory of Viticulture and Enology, College of Food Science and Engineering, Gansu Agricultural University, Lanzhou 730070, China; sameudes0@gmail.com (F.E.S.); matz@gsau.edu.cn (T.M.); Llsl07052021@163.com (Y.L.); qwl11040224@163.com (W.Q.); 2Department of Food Science and Technology, University for Development Studies, Nyankpala Campus, P.O. Box TL1882, Tamale 34983, Ghana; ratuna@uds.edu.gh (R.A.A.); fkamagloh@uds.edu.gh (F.K.A.); 3Food Technology Department, Technical College of Agricultural Engineers, Technical University of Madrid, Avenida Complutense S/N, 28040 Madrid, Spain; antonio.morata@upm.es

**Keywords:** alcohol-free wine, chemical parameters, dealcoholization, membrane, non-alcoholic wine reverse osmosis, sensory characteristics, vacuum distillation, volatile compounds

## Abstract

Over the last few years, the dealcoholization of wine has piqued the interest of winemakers and researchers. Physical dealcoholization methods are increasingly used in the dealcoholization of wines because they can partially or completely reduce the alcohol content of wines. This study aimed to compare the chemical parameters, volatile composition and sensory quality of white, rosé and red wines dealcoholized by two physical dealcoholization methods: reverse osmosis (RO) and vacuum distillation (VD) at 0.7% *v*/*v* ethanol. RO and VD effectively reduced the ethanol concentration in all wines to the required 0.7% *v*/*v*, but also significantly affected most chemical parameters. The pH, free sulfur dioxide, total sulfur dioxide, and volatile acidity decreased significantly due to dealcoholization by RO and VD, while reducing sugars and total acidity increased significantly. VD resulted in higher color intensity, which was perceptible in dealcoholized rosé and red wines, while RO caused notable color differences in dealcoholized white and red wine fractions. RO were richer in esters (more ethyl esters and isoamyl acetate), higher alcohols, organic acids, terpenics and C_13_-norisoprenoids, and carbonyl compounds, while wines dealcoholized with VD had lower levels of these volatile compounds, which may reflect both the loss of esters into the distillate during evaporation and condensation (in the case of VD) and a shift in the chemical equilibrium responsible for ester formation and hydrolysis after ethanol removal. β-damascenone exhibited the highest OAV in all wines, however, losses equal to 35.54–61.98% in RO dealcoholized fractions and 93.62% to 97.39% in VD dealcoholized fractions were observed compared to the control wines. The predominant aroma series in the original and dealcoholized wines were fruity and floral but were greatly affected by VD. Sensory evaluation and PCA showed that dealcoholization by RO improved the fruity and floral notes (in rosé and red wines), color intensity, sweetness, viscosity, and aroma intensity better than dealcoholization by VD, while VD mainly enhanced the color of the dealcoholized wines. Both methods increased the acidity of the respective dealcoholized wines. Nevertheless, RO dealcoholized wines achieved higher acceptance by the panelists than VD dealcoholized wines. Therefore, RO may be a better method for producing dealcoholized (0.7% *v*/*v*) wines with minimal impact on aroma and sensory quality.

## 1. Introduction

In recent years there has been an increasing trend of a part of consumers who want wines with lower alcoholic strength. Moreover, there is also a widespread global belief that consumption of alcoholic wines should be reduced in favor of dealcoholized, low and non-alcoholic wines [1,2,3,4]. In particular, social demands related to road safety, and some health reasons have considerably increased the demand for dealcoholized wines [5]. Interestingly, reducing the alcohol content of wines may reduce the risks associated with the consumption of alcoholic wines without compromising their cardioprotective effects [6]. The global non-alcoholic wine market is expected to grow at a remarkable compound annual growth rate (CAGR) of over 7% during the forecast period (2019–2027) and will be valued at over US$10 billion [7]. Amidst this competition, it makes sense that wine producers and researchers are trying to improve technology, especially, physical dealcoholization methods to cater to the development of non-alcoholic wines in the wine industry, considering the growing trend of consumers for wines with a lower alcoholic strength.

However, the dealcoholization of wine has two major drawbacks. The first is technical since it is extremely difficult to dealcoholize a wine while maintaining its original organoleptic characteristics. Generally, both the aromatic qualities and taste of dealcoholized wine tend to worsen the greater the decrease in its alcoholic strength [8,9,10]. The second drawback is legal since the laws vary according to each country. The International Wine Organization (OIV) establishes that a percentage of alcohol greater than 2% *v*/*v* cannot be reduced. Partial dealcoholization (up to 2% *v*/*v*) has become another practice in oenology [11]. Dealcoholization methods generate changes in the organoleptic properties of wines (even membrane methods cause compositional changes) that go beyond alcohol extraction [12,13,14,15,16,17]. This makes the resulting wines have a different flavor and aroma. However, from some scientific publications [10,18,19,20,21,22,23], good results have been achieved with dealcoholization of up to 2% *v*/*v* ethanol.

Dealcoholization methods such as osmotic distillation [16,19,24], reverse osmosis [23,25,26,27,28], pervaporation [29,30], vacuum distillation [13,14,31], and the spinning cone column [26,32] are most commonly used because they can partially or completely reduce the alcohol content (<0.5% *v*/*v*) of wines and beverages [15,17,33,34]. Dealcoholization methods, especially reverse osmosis and vacuum distillation, have been shown to have minimal impact on wine aroma and sensory quality after dealcoholization [35,36]. According to Bui et al. [37], reduced-alcohol wines produced by reverse osmosis generally have the same taste and aromas as regular wines produced by other methods such as distillation, spinning cone technology, or partial fermentation. For example, reverse osmosis was used to achieve a 75% ethanol reduction in a cider (8% *v*/*v*) without significant loss of key aroma compounds [38]. In addition, reverse osmosis was reported to retain important aroma compounds due to lower processing temperatures and better separation of compounds [39,40,41]. Vacuum distillation, on the other hand, is suitable for adjusting the alcohol content of wines [13,14,31,42]. It can also be used to extract almost all of the alcohol from a wine sample or to separate it from other less volatile components. Vacuum distillation could produce a greater reduction in alcohol and a higher concentration of other compounds such as flavonoids, organic acids, anthocyanins, and cations [13]. The ability of reverse osmosis and vacuum distillation to retain the phenolic components, aroma compounds, and sensory quality of wines after certain limits of dealcoholization make them suitable for the production of non-alcoholic or alcohol-free wines and beverages.

As far as we know, there was only one study comparing reverse osmosis and vacuum distillation in the obtainment of dealcoholized (at 5%) wine fractions at the time this experiment was conducted [13]. These authors focused on the influence of the two techniques on the physicochemical and volatile composition of the dealcoholized fractions. There is a paucity of work in the literature comparing the two dealcoholization techniques in the case of a higher level of dealcoholization (final alcohol content 0.7% by volume) of wine, as well as their effects on the sensory quality of the dealcoholized wine fractions. The latter could be used to select the best dealcoholization technique for higher dealcoholization (0.7% by volume) of wine with better preservation of important aroma compounds and sensory quality.

Therefore, in this present study, the physicochemical parameters, the composition of the volatile components and the sensory characteristics of a white, a rosé and a red wine dealcoholized by two methods reverse osmosis and vacuum distillation, were compared. The final objective of this work is to determine the most suitable method for the dealcoholization of wines.

## 2. Materials and Methods

### 2.1. Reagents and Standards

Sodium chloride and the internal standards (gas chromatographic grade) used for the quantification of volatile compounds including ethyl acetate (CAS No. 141-78-6, purity ≥ 99.8%), isoamyl acetate (CAS No. 123-92-2, purity ≥ 99.8%), 2-methylbutyl acetate (CAS No. 624-41-9, purity ≥ 99.8%), ethyl lactate (CAS No. 687-47-8, purity ≥ 99.8%), ethyl hex-anoate (CAS No. 123-66-0, purity ≥ 99.5%), ethyl octanoate (CAS No. 106-32-1, purity ≥ 99.8%), diethyl succinate (CAS No. 123-25-1, purity ≥ 99.6%), phenethyl acetate (CAS No. 103-45-7, purity ≥ 99.6%), ethyl decanoate (CAS No. 110-38-3, purity ≥ 99.8%), ethyl nona-noate (CAS No. 123-29-5, purity ≥ 99.8%), ethyl dodecanoate (CAS No. 106-33-2, purity ≥ 99.7%), ethyl hexadecanoate (CAS No. 628-97-7, purity ≥ 99.8%), isobutanol (CAS No. 78-83-1, purity ≥ 99.8%), cis-3-hexen-1-ol (CAS No. 928-96-1, purity ≥ 99.6%), 1-hexanol (CAS No. 111-27-3, purity ≥ 99.7%), 1-heptanol (CAS No. 111-70-6, purity ≥ 99.7%), 1-octanol (CAS No. 111-87-5, purity ≥ 99.8%), phenethyl alcohol (CAS No. 60-12-8, purity ≥ 99.6%), dodecanol (CAS No. 112-53-8, purity ≥ 99.5%), butanoic acid (CAS No. 107-92-6, purity ≥ 99.8%), hexanoic acid (CAS No. 142-62-1, purity ≥ 99.5%), octanoic acid (CAS No. 124-07-2, purity ≥ 99.6%), linalool (CAS No. 78-70-6, purity ≥ 99.5%), geraniol (CAS No. 106-24-1, purity≥ 99.8%), nerol (CAS No. 106-25-2, purity ≥ 99.6%), citronellol (CAS No. 106-22-9, purity ≥ 99.8%), β-damascenone (CAS No. 23696-85-7, purity ≥ 99.8%), nerolidol (CAS No. 7212-44-4, purity), 2-octanol (CAS No. 123-96-6, purity), and benzaldehyde (CAS No. 100-52-7, purity ≥ 99.6%)were purchased from Sigma-Aldrich (Shanghai, China). Deionized water (<18 MW resistance) was prepared using a Milli-Q purification system (Molecular, Chongqing, China). All reagents were of analytical purity.

### 2.2. The Wines

Three commercial wines were used in this study. A Merlot red wine (13.9% *v*/*v*) and a Pinot Noir rosé wine (12.2% *v*/*v*), both of 2020 vintage were obtained from Guo Feng winery in Gansu Province, China. A Chardonnay white wine (13.4% *v*/*v*) (2019 vintage) supplied by Mogao winery in Gansu Province, China. For each wine, 5 L was dealcoholized. After dealcoholization, all trials were bottled in 750 mL brown glass bottles (crown cork) and stored in the refrigerator (4 °C) until all experiments were performed (within 10 days).

### 2.3. Dealcoholization with Reverse Osmosis (RO)

An industrial-scale RO plant (Hangzhou Ruina Membrane Engineering Co., Ltd., Hangzhou, China) was used for dealcoholization (i.e., reducing alcohol content to 0.7% *v*/*v*) of red, rosé, and white wines according to the manufacturer’s operating instructions. The RO plant was made of stainless steel and its membrane module consisted of six spiral wound Alfa Laval RO98pHt M20 composite membranes (effective membrane area of 155 cm2, dimensions of 11.5 cm length and 300 µm diameter, pH range of 2–11, operating temperature of 5–60 °C, maximum pressure of 5.5 MPa, and salt rejection of ≥98%). Figure 1 shows the scheme of dealcoholization using the RO plant. Each wine (red, rose, and white) was dealcoholized at a constant pressure of 3.5 MPa and 20 °C [41]. Five liters of each type of wine were dealcoholized. The wine which served as feed in the feed tank (1—Figure 1) was pumped using a conveyor pump (2) to the RO membrane (3) in a tangent manner at a flow rate of 70 mL/min. In this way, the alcohol and water in the wine were able to overcome the natural osmotic pressure due to the exerted pressure, and penetrated to the permeable side of the membrane where they were collected in the permeate tank (4), while the aromatic components and flavor compounds in the wine were retained and returned to the feed tank. The flow rate of the feed and permeate was controlled by on/off valves (6), and the flow rate was measured with a rotameter (5). A manometer (7) was used to measure the feed pressure while the temperature of the feed and retentate was measured with temperature sensors (8). Since water was also removed with the ethanol, the RO process was operated in a diafiltration mode in which the volume of wine in the feed tank (1) was kept nearly constant by adding distilled water to the wine in the feed tank every hour. This allowed the continuous reduction of the ethanol content of the wine in the feed tank. After 3 h, the retentate (3.8 L of dealcoholized wine) attained an alcoholic content of 0.7% *v*/*v* and was transferred into 750 mL brown glass bottles (crown cork) and stored in a refrigerator at 4 °C until all analyses were performed within 10 days.

### 2.4. Dealcoholization with Vacuum Distillastion (VD)

A laboratory-scale distillation plant, RE-6000A rotary evaporator (Shanghai Yarong Biochemical Instrument Factory, Shanghai, China) with vacuum pump V-700, vacuum controller V-850, diagonal condenser, and water bath system (10 L volume) was used for the dealcoholization of wines. Five liters each of red, rosé, and white wines were used for dealcoholization. During dealcoholization, 400 mL was processed at a time in a 1 L flask of the vacuum distillation apparatus. The flask was covered with black plastic material to avoid light oxidation of wine components. The rotation of the flask was set at 100 rpm, the water bath temperature at 35 °C, the vacuum pressure at 0.08 MPa, and the condensation temperature at 9 ± 2 °C and was kept constant throughout all dealcoholization experiments. Each dealcoholization process was completed in 2.5 h. The water batch was replenished to the same volume and once the batch reached the set temperature, the experiment began at the speed, temperature, and pressure indicated above. At the end of each dealcoholization process, the residual wine (0.7% *v*/*v* ethanol) was transferred into 750 mL brown glass bottles (crown corked), allowed to cool at room temperature, and stored at 4 °C for further analyses (completed within 10 days).

### 2.5. Physicochemical Analyses

Physicochemical parameters of all wines and dealcoholized fractions, including ethanol, density, pH, total acidity (expressed as g/L tartaric acid), volatile acidity (expressed as g/L acetic acid), glycerol, residual sugar, and free and total sulfur dioxide (SO_2_) were all measured using a WineScan™ SO_2_ analyzer (FOSS Analytical A/S, Denmark). Color parameters including, wine color, hue, and CIELab were determined by spectral measurements with a spectrometer between 380 and 780 nm (in an interval of 2 nm), using a glass cuvette with a path length of 1 cm for rosé and white wines and a cuvette with a path length of 0.2 cm for red wine after centrifugation [42]. CIELab measurements included brightness (*L**), the intensity of red and green hues (*a**), and the intensity of blue and yellow hues (*b**). The total color difference (∆*E**) was also calculated using the following equation: ∆*E** = [(∆*L**)^2^ + (∆*a**)^2^ + (∆*b**)^2^]^1/2^ [43]. All spectrophotometric measurements were performed using a Genesis 10S UV–vis spectrophotometer (Thermo Fisher Scientific, Waltham, MA, USA).

### 2.6. Determination of Volatile Compounds

A Gas Chromatography-Mass Spectrometry (GC–MS) system (TRACE 1310-ISQ, Thermo Fisher Scientific, San Jose, CA, USA) equipped with a DB-WAX UI capillary column (60 m × 0.25 mm × 0.25 μm film thickness, Agilent Technologies) was used for the determination of volatile compounds according to the methods described by other authors [41,44,45] with slight modifications. Volatile aroma compounds from wines and dealcoholized fractions were extracted by solid-phase microextraction (SPME). Five milliliters of each sample (wines and dealcoholized fractions), 10 µL of 2-octanol (with concentration 88.2 mg/L), and 1.5 g of sodium chloride (NaCl) were put in a 15 mL glass headspace vial and immediately sealed with a polytetrafluoroethylene silicone septum. Each vial was equilibrated with stirring on a magnetic stirrer (300 rpm) at 40 °C for 30 min. Then, an SPME fiber coated with divinylbenzene/carboxen/polydimethylsiloxane (DVB/CAR/PDMS) sorbent (50/30 µm, fiber length 1 cm, Supelco, Bellefonte, PA, USA), held in a manual holder was used for extraction (30 min) of the volatile compounds in the headspace of the vial under the above conditions. The SPME fiber was then transferred to the injector port of the GC–MS system, where the volatiles were desorbed at 250 °C for 10 min in splitless mode. The GC oven temperature was started at 40 °C for 5 min, then increased to 200 °C at 4 °C/min and maintained for 20 min with helium (purity 99.9%) as the carrier gas at a constant flow rate of 1.0 mL/min. The mass spectrometer used electron ionization at 70 eV with an ion source temperature of 250 °C and a transfer temperature of 200 °C and scanned in a mass range of 50 to 350 *m/z*. The volatile compounds (obtained as peaks and mass spectra) were identified by comparing their mass spectra with those in the National Institute for Standards and Technology library (NIST 14, search version 2.0). The retention index for each compound was calculated using C_6_–C_24_ n-alkane series (Supelco, Bellefonte, PA, USA) and compared the values reported in the literature and the NIST database (http://webbook.nist.gov/chemistry/cas-ser.html (accessed on 15 August 2021). Identified volatile compounds were then quantified against the respective labelled internal standards (reported in Section 2.1). Concentrations of other volatile compounds were calculated using 2-octanol and expressed as µg/L.

### 2.7. Odour Activity Values and Aroma Series

Odor activity value (OAV) is defined as the ratio of the concentration of each volatile compound above the threshold of perception [46,47]. To determine the contribution of the identified volatile compounds to the overall aroma of the corresponding wines (white, rosé, and red), the OAV of each identified volatile compound was calculated [44,45,48]. A volatile compound with an OAV greater than 1 may contribute significantly to the overall aroma of the wine. The overall aroma of wine can be grouped into 13 aroma series including (1) fruity, (2) floral, (3) fatty, (4) spicy, (5) herbaceous/vegetative, (6) earthy, (7) chemical, (8) nutty, (9) pungent, (10) caramel, (11) woody, (12) oxidized, and (13) microbiological [47]. The aroma series for each wine type was calculated by summing the OAVs of the identified volatile compounds (OAV ≥ 0.1) in different aromat series.

### 2.8. Sensory Analysis

Sensory evaluation of the original wines and their corresponding dealcoholized fractions was performed as described by Liguori et al. [12] in a standard wine tasting laboratory of the College of Food Science and Engineering, Gansu Agricultural University. An 8-member panel comprising members of staff (3 certified in wine tasting) and students from the Department of Viticulture and Enology of the College of Food Science and Engineering, Gansu Agricultural University. Among the panelists, 5 were males and 3 were females. In addition, 4 of the panelists were teetotalers and the other 4 were alcohol-drinkers. This was to ensure the biasness of one side was eliminated, as well as to acquire an assessment by possible future consumers of the products (dealcoholized wines). The tasting was conducted between 9:30 a.m. to 12:00 p.m. A total of 9 samples including original and dealcoholized white, rose, and red wines were evaluated. A constant volume of 30 mL of each sample was served in 215 mL ISO standard (ISO 3591:1977) odor-free tasting glasses with random three-digit codes and presented randomly to the panelists. The panelists rated the wine samples at room temperature (approximately 25 °C) using a 9-point unstructured scale with scores ranging from 1 (extremely low) to 9 (extremely high) based on selected attributes including color intensity, aroma intensity, fruity and floral, red fruits, astringency, bitterness, sweetness, acidity, acidity, hotness, wine body, and overall acceptability. For original white and dealcoholized wine samples, astringency and bitterness were excluded in the evaluation. The final score was calculated for each wine as the sum of an average score of each attribute [44,49].

### 2.9. Statistical Analysis

All data were organized in Microsoft excel 2016 and analyzed using analysis of variance (ANOVA) in XLSTAT Software version 2016 (Addinsoft, New York, NY, USA). Significance was judged by Tukey’s range test at *p* < 0.05. Additionally, XLSTAT 2016 was used to carry out principal component analysis (PCA) on volatile compounds with OAV >1 and sensory attributes of wine samples. Sensory data was also analysed using Kruskal-Wallis non-parametric procedure.

## 3. Results

### 3.1. Effect of Dealcoholization on the Physicochemical Parameters

The alcohol content of Chardonnay white wine, Pinot Noir rosé wine, and Merlot red wine was reduced using RO and VD. Table 1 shows the physicochemical properties of the original wines (white, rosé, and red) and the corresponding dealcoholized fractions (0.7% *v*/*v* alcohol). There were differences in alcohol content and other physicochemical parameters of the tested wines, including amounts of reducing sugars, free sulfur dioxide, total sulfur dioxide, pH, total acidity, volatile acidity, color intensity, hue, CIELab, and glycerol. For most parameters, the dealcoholized fractions showed significant differences from their original wines (Table 1). As expected, ethanol concentration was reduced to 0.7% *v*/*v* in wines dealcoholized with RO and VD, resulting in a slight but non-significant increase in density. Compared to the original wines, RO and VD significantly increased the content of reducing sugars in the dealcoholized fractions. In particular, VD led to slightly higher sugar concentrations in white and red wines than RO, and vice versa in rose wines. In addition, VD led to increased total acidity compared to the original and RO treated wines. Similar results were reported by Motta et al. [13] who investigated the effect of distillation under vacuum and membrane contactor on the physicochemical composition and aromatic profile of dealcoholized (5% *v*/*v* ethanol) Barbera, Pelaverga, and Rosè wines.

Furthermore, RO and VD significantly decreased the concentrations of volatile acidity, free sulfur dioxide, total sulfur dioxide, and pH in the dealcoholized wine fractions. VD dealcoholized fractions had the lowest concentration of free sulfur dioxide. Given the importance of free sulfur dioxide in preventing oxidation and microbiological instability in wine, this was a very important result, highlighting the importance of monitoring sulfur dioxide levels after the dealcoholization of wines with RO or VD. In one study, free sulfur dioxide was retained in both RO and VD dealcoholized fractions (5% *v*/*v*), although it was present at lower concentrations in the original wines before dealcoholization [13]. In contrast, free sulfur dioxide was no longer found in red wines after dealcoholization (reduction of ethanol up to 2.6% abv) of Barossa Valley Shiraz Cabernet Sauvignon (14.1% abv) and McLaren Vale Cabernet Sauvignon (17.1% abv) wines by reverse osmosis-evaporative perstraction (RO-EP) [50]. The differences between these studies could be due to the different dealcoholization methods and the degree of alcohol reduction. As for glycerol content, the concentration was inversely related to ethanol loss in RO and VD dealcoholized white and rosé wine fractions, while the concentration was lower in red wine after dealcoholization by VD.

As regards wine color, both dealcoholization methods had a significant effect on the intensity of dealcoholized fractions. In particular, dealcoholization with VD significantly intensified the color of all wines compared to RO treated wines and the original wines, while the hue decreased. This could be due to the lower pH and alcohol content in this study and possibly to the higher total anthocyanin content [13]. In the same way, Motta et al. [13] found a higher color intensity of dealcoholized (5% *v*/*v*) Rosé, Pelaverga, and Barbera wines by vacuum distillation compared to RO treated wines and original wines. In addition, an Aglianico red wine with 6.5%, 10%, 11.8%, 12.3% and 12.9% *v*/*v* ethanol removal showed significant differences in color intensity and hue [16], while removal of 2% ethanol from the wine by reverse osmosis and nanofiltration resulted in increased wine color intensity of 11% and 6%, respectively [51]. Later, Russo et al. [52] and Pham et al. [50] reported a significant increase in color intensity with ethanol reduction from 13.2% to 9% *v*/*v* in Montepulciano d’Abruzzo red wine and from 14.1% to 12.5% *v*/*v* in Cabernet Sauvignon red wine by RO-EP, respectively.

In addition, CIELab measurements such as lightness (*L**), the intensity of red and green hues (*a**), and the intensity of blue and yellow hues (*b**) revealed significant differences between all original wines and their corresponding dealcoholized fractions obtained by RO and VD. Dealcoholization by VD significantly decreased the lightness (*L**) of white, rosé, and red wines by 1.6%, 11.1% and 6.3%, respectively, while red and green hues (*a**) increased by 22.6%, 23.5% and 16.26%, respectively. In contrast, RO did not affect the lightness of the rosé wine compared to the original wine but significantly decreased the lightness in dealcoholized white and red wine fractions by 1.3% and 3.3%, respectively. Compared to the original wines, RO resulted in a more significant reduction in color intensities (*a** and *b**) for all wines except red wine. According to González-Manzano et al. [53], color changes are expected to be perceptible to the human eye when ∆*E** ≥ 3.0. Dealcoholization by RO and VD had a significant effect on the color of dealcoholized red wines (∆*E** = 3.14 and 4.59, respectively, Table 1), which was detectable to the human eye. In white wine, RO also visibly affected the color (∆*E** = 3.23), which was perceptible to the human eye, while in rose wine the color difference was only noticeable in VD dealcoholized fractions (∆*E** = 7.14). In a study of partial dealcoholization of Shiraz Cabernet Sauvignon and Shiraz wines by RO-EP, a perceptible color difference (∆*E** = 5.3 and 3.0, respectively) was observed [50]. Likely, the changes in wine color characteristics observed in this study between treatments are due to more than just the concentration of ethanol remaining after dealcoholization by the two methods, but the dealcoholization process could have affected the co-pigmentation content and/or the formation of derived pigments (such as anthocyanins) as a result of oxygen uptake, sulfur dioxide loss, and/or adsorption on the membrane surface, as suggested by Gambuti et al. [54].

### 3.2. Effect of Dealcoholization on Volatile Compounds

The aroma and taste of wine are influenced by the composition of volatile compounds [45,55]. More than 1000 volatile compounds of various chemical classifications (alcohols, esters, fatty acids, aldehydes, terpenes, ketones, and sulfur compounds) are found in wine, and about 400 volatile compounds are formed during wine fermentation [56]. The removal of alcohol from wine is generally accompanied by the removal of water and some volatile compounds during dealcoholization [17]. Table 2 shows the main volatile compounds and their concentrations found in all original and dealcoholized wines. For better visualization, all the identified volatile compounds were classified into five groups: esters, alcohols, acids, terpenics and C13-norisoprenoids, and others (carbonyl compounds, aldehydes, and volatile phenols). A total of 57 different volatile compounds were identified and quantified in all wines, including 23 esters, 16 higher alcohols, 8 organic acids, 8 terpenics and C13-norisoprenoids, and 2 others.

#### 3.2.1. Esters

Many of the essential fruity aromas and flavors found in wine are imparted primarily by esters [70,71]. Esters also contribute greatly to the acid-ethyl ester balance, primarily through esterification and hydrolysis reactions during dealcoholization [50]. Consequently, the removal of ethanol from wine could affect the concentrations of esters (especially ethyl esters) and their respective acids, as the equilibrium simultaneously shifts towards ester hydrolysis Equations (1) and (2), with different equilibrium points forming for different esters [72]. Esterification and hydrolysis are reversible reactions [57], with temperature and activation energy influencing the rate of each reaction [73]. Therefore, an optimal operating temperature during dealcoholization is crucial for the acid-ethyl ester equilibrium.
Carboxylic acid + Ethanol ⇌ Ester + Water(1)
RCOOH + C_2_H_5_OH ⇌ RCOOC_2_H_5_ + H_2_O(2)

Regarding the identified esters, ethyl hexanoate, ethyl octanoate, and ethyl decanoate originate from fermentation, while diethyl succinate and ethyl lactate can be formed by chemical esterification during alcoholic fermentation and storage. The original wines had the highest total ester levels in terms of both number and concentration (ranging from 6536 µg/L to 15,654 µg/L), while the dealcoholized fractions had the lowest levels (Table 2). Esters are often hydrophobic, so losses were expected during the dealcoholization experiments, which varied depending on their physical and chemical properties. Dealcoholization by VD significantly reduced the total concentration of esters in white, rosé, and red wines by 96%, 98% and 96%, respectively, compared to the original wines, while RO resulted in 92%, 81% and 87% losses, respectively. Specifically, dealcoholization by VD resulted in significant losses in the concentrations of phenethyl acetate (77–100%), ethyl 2-hexenoate (96–100%), ethyl acetate (91–100%), ethyl decanoate (99–100%), ethyl hexanoate (99–100%), ethyl nonanoate (97–100%), and ethyl octanoate (99%) in dealcoholized wines compared to the original wines and, to a greater extent, complete losses (100%) of ethyl nonanoate, ethyl (Z) hex-3-enoate, ethyl butanoate, ethyl hexadecanoate, ethyl heptanoate, ethyl dodecanoate, and isoamyl octanoate in white or red wines (Table 2). This decrease in ester concentrations is consistent with other authors’ reports of dealcoholized white wines [12,74] and red wines [8,16,19,51,55]. In contrast, significant losses of ethyl esters of hexanoic acid, octanoic acid, and decanoic acid were observed after dealcoholization with 5% *v*/*v* ethanol using membrane contactor technology, with losses of 75–88%, 47–80%, and 34–65%, respectively [13]. RO completely (100%) removed ethyl heptanoate, ethyl nonanoate, heptyl acetate, and methyl salicylate from rose wine, while hexyl acetate was completely removed from red wine. However, the losses of esters such as ethyl hexadecanoate, ethyl acetate, hexyl acetate, and diethyl succinate were much lower in RO dealcoholized white wine fractions. A decrease in ethyl ester concentrations was also observed when red wine was partially dealcoholized by RO-EP, with higher molecular weight ethyl esters, such as ethyl octanoate and ethyl decanoate, decreasing the most, although smaller and more volatile esters were expected to permeate the RO membrane [50]. The significant losses in esters may not be solely due to their hydrophobic nature, as suggested by Diban et al. [22] and Liguori et al. [16], or by membrane filtration, but the removal of ethanol may also have affected their ester equilibrium (Equation (1)) through esterification and hydrolysis reactions, such that the esters underwent hydrolysis to release ethanol and their corresponding acids [50].

Previous studies have found that esters with higher molecular weights are hydrolyzed more rapidly [57]. Partial dealcoholization of red wine using RO-EP resulted in a decrease in ethyl ester concentrations, with higher molecular weight esters (ethyl octanoate and ethyl decanoate) decreasing more than smaller and more volatile esters such as ethyl propanoate and ethyl butanoate, which was attributed to ethanol reduction, membrane filtration, and shift in ester equilibrium [50]. In addition, the concentrations of straight-chain ethyl esters such as ethyl hexanoate, octanoate, and decanoate were found to decrease during wine ageing, which was attributed to their enzymatic production exceeding their equilibrium concentrations during fermentation [75]. Further investigation of tartaric acid esterification with ethanol in a model wine showed the influence of ethanol concentration, with higher ethanol concentrations increasing both the rate and amount of ester formation [76]. Moreover, esterification rates were found to decrease with increasing molecular weight, in the following order: ethyl decanoate > ethyl octanoate > ethyl hexanoate [73]. Subsequently, when a polypropylene hollow fiber membrane contactor apparatus was used to remove 2%, 3% and 5% *v*/*v* ethanol from two red wines with different initial ethanol concentrations (13.3% and 15.4% *v*/*v*), losses equal to 11–100% were observed in ethyl ester concentration [8]. Saha et al. [77] also found a decrease up to 80% in ethyl ester concentration when the ethanol concentration of Shiraz wine (with 13.7% *v*/*v* ethanol) and Chardonnay wine (with 12.2% *v*/*v* ethanol) decreased by up to 5% *v*/*v* during dealcoholization with a benchtop system RO-EP. Similar results were recently observed by Pham et al. [50] after dealcoholization of Shiraz Cabernet Sauvignon, Cabernet Sauvignon, and Shiraz wines by 1.6%, 2.6% and 0.7% *v*/*v* ethanol, respectively, using the reverse RO-EP. Interestingly, in these studies, the concentrations of some acids such as propanoic, decanoic, and hexanoic acids increased by 11% to 173% [77], 6% to 25% [50], and 24% [8], respectively, which may indicate ester hydrolysis.

Although both RO and VD resulted in significant ester losses, our results show that RO retains more esters than VD, which could be due to the ability of RO membrane to retain low molecular weight compounds due to its small pore size and molecular weight cutoff (MWCO) value, although this may result in severe membrane fouling and higher energy consumption [78,79]. Moreover, the temperature used in VD (35 °C) might have led to a greater loss of esters, as high temperature causes higher evaporation and thermal degradation of volatiles and other wine components. However, the retention of aroma compounds is also influenced by other factors, such as membrane properties, molecular weight, volatility and activity coefficient of each compound [80], and the non-volatile matrix of the wine [17]. Further processes can be used to recover esters lost during dealcoholization by RO and VD. In the case of VD, esters can be recovered in cryogenic traps and later added back to the wine to preserve the aroma quality, whereas, in RO, esters lost may be recovered through a second membrane filtration process such as nanofiltration or osmotic distillation and added back to the original wine in a continuous configuration.

#### 3.2.2. Higher Alcohols

Higher alcohols are produced as by-products of alcoholic yeast fermentation. They are produced mainly by the Ehrlich pathway, in which amino acids are used as substrates for transamination and decarboxylation before being reduced by alcohol dehydrogenase. In addition, yeasts can also produce higher alcohols from carbon fragments of sugar metabolism [81]. Higher alcohols at concentrations less than 300 mg/L (300,000 µg/L) contribute to the desirable wine aroma, but at concentrations greater than 400 mg/L (400,000 µg/L), a detrimental effect was observed [82,83]. In this study, the total concentration of higher alcohols in all original and dealcoholized wines ranged from 199 µg/L to 9705 µg/L, with no concentration exceeding 400,000 µg/L, so no undesirable aroma was perceived in both original and dealcoholized wines. However, the degree of dealcoholization by RO or VD was shown to affect the loss of higher alcohols in the wine. The removal of 4.2% ethanol from Montepulciano d’Abruzzo red wine (13.2% *v*/*v*) by RO resulted in a loss of 30% of the total concentration of higher alcohols [52]. Dealcoholization of a Verduno Pelaverga red wine (14.6% *v*/*v*) at 5% *v*/*v* also resulted in a loss of higher alcohol content of about 53% [13].

In the current study, where alcohol content was reduced to 0.7% *v*/*v*, the majority of higher alcohols were still present in both the original and dealcoholized wines (despite losses), with 1-pentanol being the most abundant (1 µg/L to 6306 µg/L), followed by 2-phenylethanol (185 µg/L to 2785 µg/L), but at concentrations below the odor threshold, especially in RO and VD dealcoholized wines. RO resulted in 67%, 58% and 75% losses in the total concentration of higher alcohols in white, rosé, and red wines, respectively, while VD caused about 95%, 85% and 94% losses in white, rosé, and red wines, respectively. VD resulted in the complete removal of some higher alcohols in dealcoholized white wine (including benzyl alcohol, *cis*-3-hexen-1-ol, 3-ethoxy-1-propanol, and isobutanol), dealcoholized rosé wine (i.e., 1-undecanol, 1-decanol, 2-ethyl-1-hexanol, and isobutanol), and dealcoholized red wine (including 1-decanol, 2-ethyl-1-hexanol, 1-hexanol, isobutanol, and 4-methyl-1-pentanol). On the other hand, the content of isobutanol, 3-ethoxy-1-propanol, and 1-heptanol remained almost unchanged after dealcoholization by RO (especially in dealcoholized white wine), while 2-ethyl-1-hexanol increased by 55% and 4% in RO dealcoholized rosé and red wines, respectively. In contrast, in a similar study in which Langhe Rosè wine, Barbera red wine, and Verduno Pelaverga red wine were dealcoholized at 5% *v*/*v* ethanol using the membrane contactor technique, higher losses of isobutanol (between 72% and 75%) were reported [13]. Moreover, Liguori et al. [16] observed no change in 1-hexanol concentration in wine fractions dealcoholized by 6.5% alcohol using osmotic distillation, while in this study the concentration of 1-hexanol was significantly reduced in all wines after dealcoholization by RO and VD. In agreement with our results, other studies found losses of 5% to 15% 1-hexanol in red wines dealcoholized by 2.5% alcohol by RO-EP [50], 84% losses in Chardonnay wine dealcoholized by 10.61% *v*/*v* ethanol using vacuum distillation [14], 66% to 71% losses in wines (Langhe Rosè, Barbera red, and Verduno Pelaverga wines) dealcoholized at 5% alcohol by RO and VD [13], 17% to 35% losses in white and red wines dealcoholized by 4% alcohol by a membrane contactor [74], and 28% to 38% losses in Aglianico wines dealcoholized by 5% alcohol with an osmotic distillation method [8]. In the later studies, losses of 1-hexanol increased with the degree of dealcoholization [8,74].

The finding of our study shows that RO is more effective than VD for higher alcohol retention. Nevertheless, the retention was much lower compared to the original wines, which could be due to the operating conditions as well as the vapor pressure and volatility of the different alcohol compounds [17]. Other factors such as membrane properties and molecular weight [80], as well as the aforementioned non-volatile matrix of the wine [17], might have also influenced the retention of the higher alcohols.

#### 3.2.3. Organic Acids

Organic acids are by-products of yeast metabolism, mainly produced during alcoholic fermentation or malolactic fermentation. The cheese and fat flavors often perceived in wine are closely related to the acidic compounds present in the wine [84]. In this experiment, the total organic acid content in white, rose, and red wines (original and dealcoholized) ranged from 328 µg/L to 2207 µg/L, 331 µg/L to 3659 µg/L, and 125 µg/L to 1402 µg/L, respectively. Dealcoholization by RO and VD resulted in a significant decrease in all organic acids. In white and red wines, dealcoholization by VD resulted in the highest losses of organic acids of 85% and 91%, respectively, while dealcoholization by RO resulted in a decrease in the concentration of acids in dealcoholized rose wine by up to 91%, with higher proportions of lower molecular weight acids, such as acetic and butanoic acids, decreasing more than higher molecular weight acids, such as hexanoic, octanoic, and decanoic acids. Similarly, Ivić et al. [41] found higher permeability of reverse osmosis and nanofiltration membranes to acetic acid, which they attributed to the low molecular weight of this compound (60.05 g/mol). Acetic acid is a by-product of alcoholic or lactic fermentation. When present in small amounts, it can enhance wine aroma, but in higher concentrations, it can produce a bitter and sour aftertaste or vinegar-like aroma, which is regarded as a wine fault [85]. Therefore, the ability of RO and VD to reduce acetic acid content could be used to regulate the acetic acid concentration in wines. Moreover, acetic acid is the main component of the volatile acid group in wine, which explains why the total concentration of volatile acids in all dealcoholized wines was lower than in the original wines (Table 1).

Concentrations of branched acids such as 2-methylbutanoic acid and 2-methylhexanoic acid also decreased significantly in all RO and VD dealcoholized wines, with complete loss of 2-methylbutanoic acid in RO and VD dealcoholized white wines, and in VD dealcoholized rose wines (100%). Furthermore, RO resulted in the complete loss of butanoic and isobutyric acids in dealcoholized white wine and the complete elimination of isobutyric acid in rose wine. Theoretically, the concentration of acids should increase slightly when ester hydrolysis occurs. However, the loss of acids due to dealcoholization by RO and VD caused most acid concentrations to decrease. Compared to the original wines, total organic acid losses in RO dealcoholized white, rosé, and red wines were 73%, 89% and 76%, respectively, while in VD dealcoholized white, rosé, and red wines, losses were 85%, 91% and 91%, respectively. These losses were similar to white wines [12] and red wines [8,16] dealcoholized at higher levels, but higher than Verdicchio and Soave dealcoholized at levels of −2% and −3% *v*/*v* [74], Barbera, Rosè, and Pelaverga dealcoholized at 5% *v*/*v* [13], and Shiraz Cabernet Sauvignon, Cabernet Sauvignon, and Shiraz dealcoholized at −1.6%, −2.6%, and −0.7% *v*/*v* ethanol, respectively.

#### 3.2.4. Terpenics and C_13_-Norisoprenoids

Terpenes are known to impart floral and fruity aromas to the wine. They can be produced by yeasts during alcoholic fermentation when glucosidases bind freely to glycosylated precursors [86]. The concentration of terpenics and C13-norisoprenoid compounds (citronellol, geraniol, nerol, β-damascenone, linalool, α-terpineol, and geranyl acetone) in wines followed a similar trend as esters and higher alcohols. The total concentrations of terpenes in dealcoholized and original wines were within the ranges of 7 µg/L to 112 µg/L in white wine, 5 µg/L to 63 µg/L in rosé wine, and 2 µg/L to 47 µg/L in red wine. Both RO and VD significantly reduced the concentrations of all terpenic and C13-norisoprenoid compounds after dealcoholization. Between the dealcoholized fractions of RO and VD, the highest concentrations of terpenes were found in the dealcoholized fractions obtained with RO, with β-damascenone showing the highest concentration. Furthermore, geranyl acetone, belonging to the C13-norisoprenoids, was significantly affected by RO and VD dealcoholization, with RO leading to losses between 35% and 100% and VD leading to losses between 92% and 100% in all wines. Contrarily, VD resulted in a 100% loss of citronellol, α-terpineol, and nerol in white, rosé, and red wines, respectively. Given the strong hydrophobicity of these terpenes, a greater reduction, roughly equivalent to that of esters, was to be expected due to both their chemical affinity for the membrane and their high volatility upon heating. Losses of terpenes were also reported after ethanol reduction of up to −5% *v*/*v* in red wines by a polypropylene hollow fiber membrane contactor method [8]. However, the terpene losses were lower compared to those observed in this study. The difference in terpene losses between the two studies could be due to the degree of ethanol removal, the nature of the non-volatile matrix, and/or the dealcoholization method and operating parameters used in these studies.

#### 3.2.5. Carbonyl Compounds

Only three carbonyls from ketonic, aldehydic, and sulfur compounds were found, but at low concentrations. These compounds differed significantly in all treatments (i.e., original and dealcoholized wines). RO and VD resulted in lower concentrations of all these compounds compared to their concentrations in the original wines. Benzaldehyde, which has a bitter almond taste, was detected in all wines, but at very low concentrations, especially in VD dealcoholized wines (1 g/L to 6 g/L). Methionol, characterized by a sulfurous note, disappeared completely in RO dealcoholized rose wine and in VD dealcoholized white, rosé, and red wines. Similarly, Liguori et al. [16] discovered lower levels of benzaldehyde and methionol in an Aglianico red wine the higher the degree of dealcoholization. In contrast, methionol in Langhe Rosè wine (13.2% *v*/*v*), Verduno Pelaverga red wine (15.2% *v*/*v*), and Barbera red wine (14.6% *v*/*v*) remained stable or recorded relatively low losses after dealcoholization at 5% *v*/*v* ethanol by membrane contactor and vacuum distillation [13]. The loss or removal of these volatile compounds in this study can be considered positive, as these compounds are responsible for off-flavors in wine.

In summary, the disparities in the aroma profiles of the wines dealcoholized by the two methods (RO and VD) were both qualitative and quantitative. The wines dealcoholized with RO were richer in esters (i.e., more ethyl esters and isoamyl acetate), higher alcohols, organic acids, terpenes, and carbonyl compounds, while the wines dealcoholized with VD had lower levels of these volatile compounds, demonstrating that RO was superior to VD in retaining wine desirable volatiles in this study. This is consistent with a recent study [41] reporting that RO improves retention of desirable wine constituents (such as ethyl esters and acetates) while removing compounds (including 4-ethylguaiacol) that have a negative impact on wine sensory properties. Similar to our study but contrarily to our findings, Motta et al. [13] found that dealcoholization by distillation under vacuum retained volatile compounds, especially esters, better than the membrane contactor technique. The dissimilarity between these two studies could be due to the different operating conditions, the different degree of alcohol reduction, and the fact that in their study the first fractions of the distillate from vacuum distillation were reintroduced into the final dealcoholized fraction, while this was not the case in our study. According to Motta et al. [13], the esters that are lost during distillation are often concentrated in the first fractions of the distillate. Esters, especially ethyl esters and acetates, are associated with fruity odor notes and have a low perception threshold, while terpenes are responsible for some of the main floral aromas in wines Moreover, higher alcohols, which do not directly contribute to wine aroma composition due to high perception thresholds, are crucial for the formation of the “aroma pool”, the matrix that regulates wine olfactory translation. Consequently, wines dealcoholized with RO would have better sensory quality than wines dealcoholized with VD as evident in our aroma series and sensory results shown in Figure 2 and Figure 3.

### 3.3. Odor Activity Values (OAVs)

A volatile with an OAV >1 can contribute significantly to the overall aroma of the wine [45,76,77], but it is also reported that a compound with 0.1< OAV <1 should be considered in line with the widely accepted theory that even compounds below the threshold can contribute to wine aroma via additive or synergistic influences of the volatile compounds that are part of the wine composition [47]. In this study, the OAV was calculated and used to evaluate which compounds contributed most to the overall aroma in the wine samples based on the identified and quantified volatile compounds detected in the original and dealcoholized wines. Appendix A shows the volatile compounds that could contribute significantly to the overall aroma of the white, rosé, and red wines. A total of 12 volatile compounds with an OAV >1 were found in both the original and dealcoholized wines, including six esters (ethyl decanoate, ethyl hexanoate, ethyl octanoate, hexyl acetate, isoamyl acetate, phenethyl acetate), two alcohols (1-hexanol, dodecanol), two acids (hexanoic acid, octanoic acid), and two terpenes (β-damascenone, geraniol). These compounds were found in both original and dealcoholized wines in the following order: original wines > RO dealcoholized wines > VD dealcoholized wines.

Among the volatile compounds with OAV >1, β-damascenone had the highest OAV, followed by isoamyl acetate, ethyl hexanoate, ethyl octanoate, and ethyl decanoate. No significant differences in the OAV of β-damascenone were observed in original and RO dealcoholized white wines, yet VD dealcoholized white wine was significantly different from original white wine. The OAV of β-damascenone in RO and VD dealcoholized white wines, decreased by 44% and 95%, respectively. It has been reported that β-damascenone imparts floral and fruity notes to wine [87]. Regarding isoamyl acetate, it had the highest OAV in all original wines and was reduced by 60% to 67% in RO wines (especially in RO dealcoholized red wine) and by 76% to 100% in VD wines (especially in VD dealcoholized rose wine). Other compounds with high OAVs were ethyl hexanoate and ethyl octanoate. The OAV of ethyl hexanoate also decreased by 83% to 93% in RO dealcoholized wines and by 100% in VD dealcoholized wines, while the OAV of ethyl octanoate decreased by 100% in both RO and VD wines (Appendix A). However, both compounds had OAVs >9. Isoamyl acetate, ethyl hexanoate, ethyl octanoate, and ethyl decanoate are very important for the quality of wine and contribute mainly to fruity aromas [46,48,79]. Octanoic acid and 1-hexanol also showed relatively low OAV (1 < OAV < 4) but were only detected in the original wine samples. Despite the reduction of volatile compounds with high OAVs in RO and VD dealcoholized wines, they are still considered to contribute to the aroma quality of the dealcoholized wines (floral, fruity, oily, and herbaceous notes), as their OAVs were still higher than 1.

### 3.4. Aroma Series

The wine aroma wheel can be used to classify wine aroma into several aroma series [47]. To link the chemical composition to the sensory profile, the odor descriptors of the volatile compounds in the wine samples were grouped into several aromatic series, using volatile compounds with an OAV ≥0.1. The aromatic series used to define the aroma of white, rosé, and red wines were floral, fruity, fatty, pungent, spicy, vegetative, and earthy. Due to the difficulty in identifying their odor sensations, several volatile compounds were assigned to one or more aromatic series [55,62,88]. Figure 2a–c shows the total intensity of each aromatic series used to characterize the aroma of white, rosé, and red wines (except for the earthy, pungent, and spicy aromatic series), calculated as the sum of each volatile compound assigned to each series with OAV ≥0.1. As can be seen, fruity and floral aromas were the dominant aroma series of the white, rosé, and red wines, carried mainly by some esters (such as isoamyl acetate, ethyl hexanoate, ethyl octanoate, ethyl decanoate, hexyl acetate, and phenethyl acetate) and terpenes (i.e., β-damascenone, linalool, citronellol, and geranyl acetone) (Appendix A). Although all the wine samples tested had a similar order of aroma series, the content of each aroma series differed (Figure 1a–c), suggesting that dealcoholization played a significant role in altering the aromatic quality of the wines.

Both the RO and VD dealcoholization methods affected all aroma series of all wines, with VD dealcoholization showing the greatest reduction in all aroma series than RO. The reduction of the most dominant aroma series (fruity) in all RO dealcoholized wines (white, rosé, and red) varied from 55% to 67%, while the floral aroma series decreased by 4% to 62%, the fatty aroma series by 67% to 89% and the vegetative aroma series by 60% to 67%. VD Dealcoholization, on the other hand, resulted in a 94% to 99% decrease in the fruity aroma series, 94% to 97% in the floral aroma series, 83% to 100% in the fatty aroma series, and 100% in the vegetative aroma series for all wines. The complete loss (no odor value) of the vegetative (in all wines) and fatty (in red wine) aroma series as a result of dealcoholization by VD indicates that these types of aromas cannot be perceived by humans. Although the fatty and vegetative aroma series were the lesser aroma series in all original, RO, and VD wine samples, they could still enhance wine aroma quality because these aromas can be mediated by some alcohols (1-hexanol and 1-octanol), fatty acids (including decanoic acid, hexanoic acid, octanoic acid) and esters (hexyl acetate). However, high levels of these aromas can negatively affect the wine aroma quality [63,64,65]. Between RO and VD, RO had the least effect on all aroma series, except for the fatty aroma series of rose wine, where both RO and VD resulted in similar levels of redaction (89%) compared to the original rose wine.

### 3.5. Effect of Dealcoholization on Sensory Characteristics

Sensory analysis was used to examine the original white, rosé, and red wines, as well as their dealcoholized fractions obtained from RO and VD, to determine whether there were noticeable differences in their sensory characteristics. As can be seen in Figure 3a–c, significant differences in sensory properties were found for all wine samples (original, RO, and VD), where samples did not show very comparable scores. With the reduction of ethanol by RO and VD in all wines, acidity, color intensity, and astringency (only in rose and red wines increased, while the perceptions of hotness, bitterness, wine body, sweetness (only in white and red wines), aroma intensity, and notes of red fruits (only in rosé and red wines) decreased significantly. This was mainly due to ethanol removal. As ethanol is the most abundant volatile compound in wines, ethanol concentration can affect the perception of wine aroma and taste, as well as various mouth and palate sensations [22,66,67,68]. Higher levels of ethanol in wine tend to enhance perceptions of astringency, body, bitterness, and hotness, while lower levels of ethanol may reduce persistence in the mouth, aroma intensity, and flavor [59,60,61,68,69,89]. Additionally, ethanol has a masking effect on acidity and astringency, as it has sweet, soft, and harmonizing notes [18]. Similar to our study, other researchers [8,12,19] also reported an increase in acidity and astringency after dealcoholization. The low sensations of red fruit and fruity and floral notes during wine tasting were unexpected, as the fruity and floral aroma series were the most important in both the original and the dealcoholized white, rosé, and red wines. This could be because the total intensity of the aroma series was calculated as the sum of all volatile compounds with OAVs ≥0.1, without considering the other compounds in the wine matrix. However, taking into account the interaction, suppression, and influence of the wine matrix, the intensity of the sensory attributes of the wine aroma profile may change so that the descriptors of some aroma series (4—pungent, 5—spicy and 7—earthy) are not perceived while the intensity of others is enhanced. Between the dealcoholized wines obtained by RO and VD, those from RO exhibited better wine body, fruity and floral notes, and higher aroma intensity. This was mainly attributed to the fact that RO retained more aroma compounds (ethyl esters and acetates, which, as mentioned earlier, introduce fruity and floral notes into the wines) better than VD.

Regarding overall acceptability, original wines scored higher (>6) for all wines, followed by RO dealcoholized wines (≥6), with VD dealcoholized wines having the least acceptable scores (≤6). However, there were no significant differences between the original wines and RO dealcoholized white and rose wines for overall acceptability. This could be because the RO wines had good scores for aroma intensity, color intensity, wine body, and fruity and floral notes. In addition, this could be due to the higher concentration of aroma compounds in the original white and rosé wines compared to the original red wines, so there were still considerable concentrations after dealcoholization by RO as shown in Table 2, which contributed to the sensory quality of RO dealcoholized wines. Based on our sensory results on RO and VD dealcoholized wines, comparatively, RO dealcoholized wines had better sensory quality and higher acceptability among panelists than VD dealcoholized wines.

### 3.6. Principal Component Analysis (PCA)

Principal component analysis (PCA) was used to investigate the possible grouping of both the original and dealcoholized white, rosé, and red wine samples according to the sensory attributes and volatile compounds with an OAV >1 that may contribute significantly to the overall aroma of the wine [44,90,91]. Figure 4 shows that the first principal component (PC1) had the greatest influence and accounted for 50.03% of the variability, while the second principal component (PC2) accounted for 28.71% of the variability so that PC1 and PC2 together explained 78.80% of the total variance. A clear distinction was found between the original wines and the dealcoholized wines (RO and VD), as the original wines were located in the positive upper (original rose and red wines) and lower (original white wine) quadrants of PC1 and the dealcoholized wines were located in the negative upper (dealcoholized rose and red wines) and lower (dealcoholized white wine) quadrants of PC1. A further distinction was made in the dealcoholized wines according to PC2, where RO and VD dealcoholized rose and red wines were clustered together on the positive part of PC2 and separated from the original wines, VD dealcoholized white wine, and RO dealcoholized white wine.

Hotness, bitterness, wine body, fruity and floral notes, red fruits, aroma intensity, sweetness, and overall acceptability were the main components contributing to the sensory quality of the original wines, while ethyl hexanoate, ethyl octanoate, hexyl acetate, isoamyl acetate, phenethyl acetate, 1-hexanol, dodecanol, hexanoic acid, octanoic acid, β-damascenone, and geraniol were the main volatile compounds contributing to the overall aroma of the wine. The clustering of all RO and VD dealcoholized wines on the opposite side for these parameters suggests, that dealcoholization by RO or VD reduced the concentration of the main volatile compounds (which is consistent with the volatile compound results in Table 2) contributing to the overall aroma of the wine, as well as the sensory quality of the dealcoholized wines (consistent with the sensory results in Figure 2) compared to their respective original wines. This also explains why these wines had lower preference scores compared to the others (Figure 1). An ethanol reduction of 5.5% *v*/*v* in wine can significantly decrease the preference. Furthermore, dealcoholized wines (RO and VD rose and red wines) were positively correlated with astringency and acidity on the negative upper quadrant of PC1, indicating the impact of ethanol removal on wines as higher levels of ethanol reduction can result in increased astringency and acidity of dealcoholized wines [8]. Similarly, in other studies, astringency and acidity were positively correlated (i.e., in the same group) with dealcoholized red wine [8] by PCA analysis. As regards RO and VD dealcoholized white wines, they were not associated with any components contributing to sensory quality or overall aroma indicating that RO or VD significantly affected the sensory characteristics and aroma compounds of these wines.

## 4. Limitations

The findings of this study are specific to wines produced from Chardonnay, Pinot Noir, and Merlot grape varieties. It is possible that the dealcoholization of wine produced from terpene varieties such as Muscats, Gewürztraminer, Riesling, and Torrontés would have a greater positive influence on the overall sensory profile of those wines as these varieties have high contents of fruity and floral varietal aroma. Therefore, further research is necessary, taking into account the terpene varieties mentioned above to understand the effects of the two different dealcoholization methods on other highly aromatic wine styles. Additionally, more consumers (especially non-alcoholic wine consumers) should be involved in the sensory evaluation of the wine samples to obtain a more representative sample. Furthermore, using only non-alcoholic wine consumers would have likely benefited the results, as it would enable an investigation into whether differences in sensory perception of the dealcoholized wine fractions were the result of alcohol reduction, or because of the loss of aroma compounds during the dealcoholization process.

## 5. Conclusions

In the present study, the physicochemical parameters, composition of volatile components, and sensory characteristics of white, rosé, and red wines dealcoholized by two methods; reverse osmosis and vacuum distillation at an alcohol content of 0.7% *v*/*v* were compared to determine which method is more suitable for dealcoholization of wines. Both RO and VD removed ethanol to a controllable extent, but also significantly affected the physicochemical parameters. From a practical perspective, wine producers using RO or VD in the dealcoholization of wine may need to consider the risks associated with lower concentration or loss of free sulfur dioxide (which can act as a preservative) or oxygen uptake (which can lead to oxidation during ageing) during dealcoholization. Between the two dealcoholization methods, RO showed better retention of volatile compounds, while VD produced the most significant reduction. Moreover, RO improved the fruity and floral notes (in rosé and red wines), sweetness, viscosity, and aroma intensity compared to dealcoholization by VD, although VD mainly improved the color of the wine. In addition, the wines from RO achieved higher acceptance by the panelists than the wines from VD. Therefore, RO could be a better method for dealcoholization of wine to minimize possible negative effects on wine quality while meeting social demands related to road safety and some health reasons which have considerably increased the demand for dealcoholized wines.

## Figures and Tables

**Figure 1 membranes-11-00957-f001:**
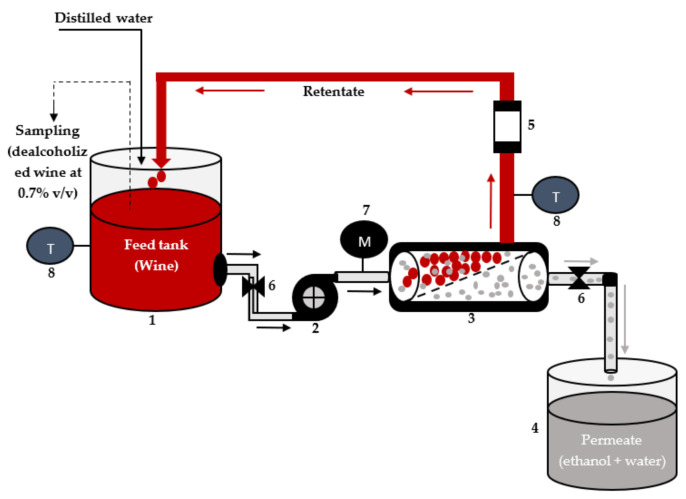
Scheme of the reverse osmosis unit used for removing alcohol from wine (1—feed tank, 2—conveyor pump, 3—membrane module, 4—permeate tank, 5—rotameter, 6—on/off valves, 7—manometer, 8—temperature sensors).

**Figure 2 membranes-11-00957-f002:**
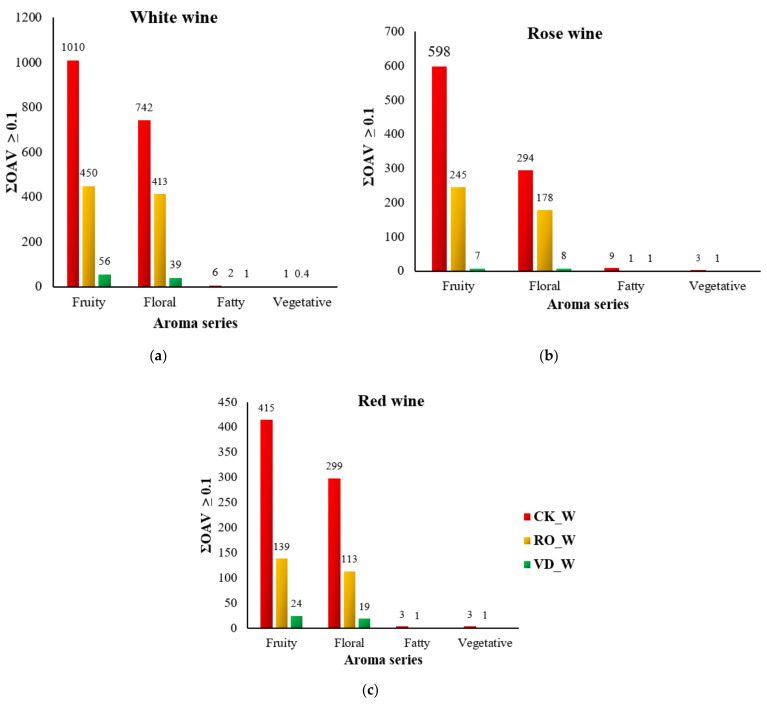
Aromatic series in (**a**) white, (**b**) rose, and (**c**) red wines (ΣOAV ≥ 0.1). CK: control; RO: reverse osmosis; and VD: vacuum distillation.

**Figure 3 membranes-11-00957-f003:**
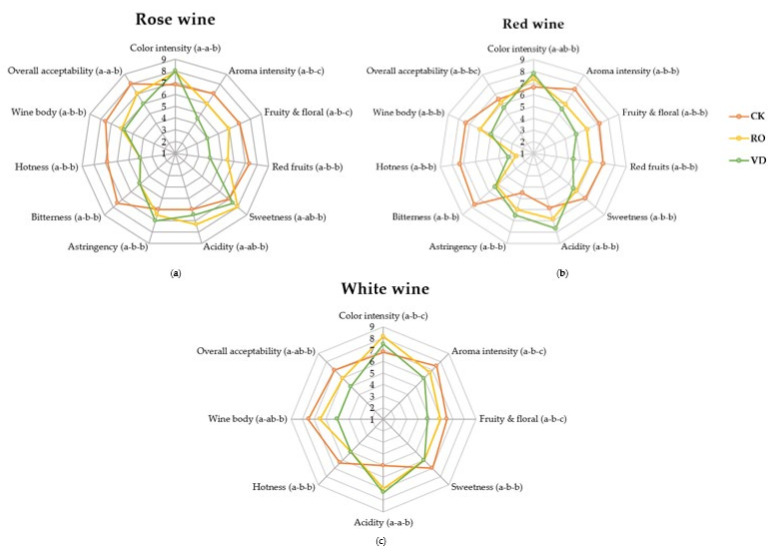
Spider plot of sensory analysis (means) for (**a**) rose wine, (**b**) red wine, and (**c**) white wine. Different letters represent significant differences at a significant level of 0.05. CK: control; RO: reverse osmosis; and VD: vacuum distillation.

**Figure 4 membranes-11-00957-f004:**
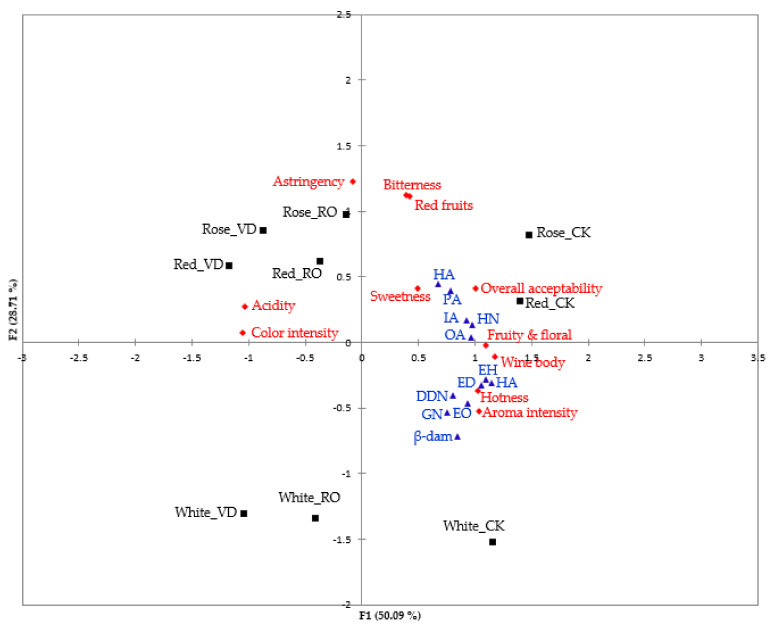
Principal component analysis (PCA) of sensory attributes and volatile compounds with an OAV >1 in wine samples with different dealcoholization methods. White_CK: original white wine, Rose_CK: original rose wine, Red_CK: original red wine, White_RO: dealcoholized white wine by reverse osmosis, Rose_RO: dealcoholized rose wine by reverse osmosis; Red_RO: dealcoholized red wine by reverse osmosis, White_VD: dealcoholized white wine by vacuum distillation, Rose_VD: dealcoholized rose wine by vacuum distillation, Red_VD: dealcoholized red wine by vacuum distillation, ED: ethyl decanoate. EH: ethyl hexanoate, EO: ethyl octanoate, HA: hexyl acetate, IA: isoamyl acetate, PA: phenethyl acetate, HN: 1-hexanol, DDN: dodecanol, HA: hexanoic acid, β-dam: β-damascenone, GN: geraniol, and OA: octanoic acid.

**Table 1 membranes-11-00957-t001:** Physicochemical composition of original and dealcoholized wine fractions obtained by the two techniques at 0.7% *v*/*v* alcohol.

Chemical Parameter	White Wine			Rosé Wine			Red Wine		
	CK	RO	VD	CK	RO	VD	CK	RO	VD
Alcohol (% *v*/*v*)	13.4 ^a^	0.7 ^b^	0.7 ^b^	12.2 ^a^	0.7 ^b^	0.7 ^b^	13.9 ^a^	0.7 ^b^	0.7 ^b^
Reducing sugars (g/L)	1.10 ^c^	2.00 ^b^	2.80 ^a^	15.90 ^c^	17.90 ^a^	16.30 ^b^	0.93 ^c^	3.13 ^b^	4.93 ^a^
Density	0.99 ^a^	1.00 ^a^	1.01 ^a^	1.00 ^a^	1.02 ^a^	1.02 ^a^	0.99 ^a^	1.01 ^a^	1.02 ^a^
Free SO_2_ (ppm)	1.00 ^a^	1.00 ^a^	0.83 ^b^	14.20 ^a^	4.80 ^b^	1.60 ^c^	10.60 ^a^	6.93 ^b^	1.30 ^c^
Total SO_2_ (ppm)	24.00 ^a^	21.90 ^b^	20.23 ^c^	25.30 ^a^	21.50 ^c^	24.23 ^b^	21.50 ^a^	14.57 ^b^	12.93 ^c^
pH	3.72 ^a^	3.55 ^c^	3.68 ^b^	3.72 ^a^	3.50 ^c^	3.62 ^b^	3.34 ^a^	3.12 ^c^	3.28 ^b^
Total acidity (g/L)	4.36 ^c^	4.71 ^b^	6.39 ^a^	6.07 ^b^	5.81 ^b^	8.27 ^a^	4.99 ^b^	4.25 ^c^	5.69 ^a^
Volatile acidity (g/L)	0.32 ^a^	0.07 ^c^	0.31 ^b^	0.29 ^a^	0.11 ^b^	0.27 ^a^	0.5 ^a^	0.09 ^c^	0.43 ^b^
Glycerol (g/L)	5.90 ^b^	2.70 ^c^	6.50 ^a^	4.50 ^b^	2.20 ^c^	5.43 ^a^	8.30 ^a^	5.57 ^b^	3.90 ^c^
Color intensity (au)	0.39 ^b^	0.30 ^c^	1.22 ^a^	1.96 ^b^	1.91 ^b^	2.35 ^a^	4.39 ^c^	7.53 ^b^	7.92 ^a^
Hue	4.46 ^a^	1.35 ^c^	2.00 ^b^	0.93 ^a^	0.78 ^b^	0.79 ^b^	0.78 ^a^	0.76 ^a^	0.68 ^b^
*L**	96.44 ^a^	95.19 ^b^	94.91 ^c^	54.21 ^a^	54.20 ^a^	48.20 ^b^	50.43 ^a^	48.79 ^b^	47.27 ^c^
*a**	−0.31 ^b^	−1.63 ^c^	−0.24 ^a^	36.53 ^b^	34.14 ^c^	45.12 ^a^	42.26 ^c^	48.79 ^b^	49.13 ^a^
*b**	13.38 ^a^	9.34 ^b^	14.94 ^a^	6.43 ^b^	5.79 ^c^	8.80 ^a^	4.00 ^c^	7.79 ^b^	7.80 ^a^
∆*E* 1*	–	3.23	1.31	–	1.00	7.14	–	3.14	4.59

Means (*n* = 3) with different letters (^a–c^) in the same row differ significantly (*p* < 0.05, Tukey’s test) from one another for the different parameters. CK: control; RO: reverse osmosis; and VD: vacuum distillation.

**Table 2 membranes-11-00957-t002:** Concentrations (µg/L) of volatile compounds present in the original wines and the fractions dealcoholized (0.7% *v*/*v* ethanol) with both methods.

Compounds	RI DB-WAX	White Wine			Rosé Wine			Red Wine			Odor Descriptor ^a^	Odor Threshold (µg/L) ^b^	Aroma Classes ^c^
		CK	RO	VD	CK	RO	VD	CK	RO	VD			
Esters													
Ethyl acetate	897	61 ^a^	41 ^b^	6 ^c^	398 ^a^	49 ^b^	3 ^c^	40 ^a^	16 ^b^	ND	Fruity, balsamic	7500	2,4
Isobutyl acetate	1019	ND	ND	ND	6 ^a^	3 ^b^	ND	ND	ND	ND	Fruity	1600	2
Ethyl butanoate	1041	38 ^a^	5 ^b^	ND	19 ^a^	9 ^b^	1 ^c^	31 ^a^	16 ^b^	ND	Floral, fruity	400	1,2
Isoamyl acetate	1128	2155 ^a^	730 ^b^	516 ^b^	5422 ^a^	1777 ^b^	ND	1122 ^a^	453 ^b^	173 ^c^	Fruity	30	2
Ethyl hexanoate	1238	2152 ^a^	184 ^b^	2 ^b^	1712 ^a^	127 ^b^	2 ^c^	966 ^a^	160 ^b^	1 ^c^	Fruity	14	2
Hexyl acetate	1277	30 ^a^	19 ^b^	9 ^c^	1157 ^a^	196 ^b^	86 ^b^	72 ^a^	ND	ND	Fruity, herb	670	2,6
Ethyl (Z)hex-3-enoate	1296	9 ^a^	1 ^b^	ND	ND	ND	ND	ND	ND	ND	/	/	/
Ethyl heptanoate	1338	1 ^a^	ND	ND	3 ^a^	ND	ND	ND	ND	ND	Fruity	220	2
Ethyl 2-hexenoate	1350	7 ^a^	1 ^b^	1 ^b^	17 ^a^	3 ^b^	ND	8 ^a^	2 ^b^	ND	/	/	/
Heptyl acetate	1377	ND	ND	ND	6 ^a^	ND	ND	ND	ND	ND	Floral	1500	1
Ethyl octanoate	1439	8991 ^a^	92 ^b^	18 ^b^	3856 ^a^	206 ^b^	17 ^c^	2505 ^a^	58 ^b^	6 ^c^	Floral, fruity	240	1,2
Ethyl trans-4-decenoate	1503	1	ND	ND	38 ^a^	8 ^b^	1 ^c^	252 ^a^	20 ^b^	ND	/	/	/
Ethyl nonanoate	1522	92 ^a^	11 ^b^	3 ^c^	3 ^a^	ND	ND	8 ^a^	ND	ND	Floral, fruity	1300	1,2
Ethyl 3-hydroxybutyrate	1523	6 ^a^	2 ^b^	2 ^b^	ND	ND	ND	10 ^a^	8 ^ab^	7 ^b^	/	20,000	2
Isoamyl lactate	1554	ND	ND	ND	3 ^a^	1 ^b^	1^c^	ND	ND	ND	/	/	/
Ethyl decanoate	1640	1819 ^a^	17 ^b^	5 ^b^	1177 ^a^	97 ^b^	2 ^c^	736 ^a^	4 ^b^	ND	Fruity	200	2
Isoamyl octanoate	1662	26 ^a^	3 ^b^	ND	8 ^a^	6 ^b^	ND	ND	ND	ND	Fruity	125	2
Diethyl succinate	1679	29 ^a^	18 ^b^	12 ^c^	9 ^a^	6 ^b^	5 ^b^	339 ^a^	97 ^b^	86 ^b^	Fruity	6000	2,3
Benzyl acetate	1736	ND	ND	ND	10 ^a^	2 ^b^	1 ^b^	ND	ND	ND	Green	/	6
Methyl salicylate	1751	ND	ND	ND	4 ^a^	ND	ND	ND	ND	ND	Peppermint	40	/
Phenethyl acetate	1823	235 ^a^	62 ^b^	1 ^b^	693 ^a^	297 ^b^	162 ^c^	87 ^a^	32 ^b^	ND	Floral	250	1
Ethyl dodecanoate	1847	ND	ND	ND	ND	ND	ND	297 ^a^	ND	ND	Floral, fruity	3500	1,2
Ethyl hexadecanoate	2245	2 ^a^	1 ^b^	ND	14 ^a^	6 ^b^	1 ^c^	3 ^a^	1 ^b^	ND	Fruity, waxy	1500	2,3
Total esters		15,654	1187	575	14,555	2793	282	6536	867	273			
Higher alcohols													
Isobutanol	1099	3 ^a^	3 ^a^	ND	3 ^a^	2 ^b^	ND	71 ^a^	18 ^b^	ND	Bitter, green	40,000	3,6
1-Pentanol	1214	2556 ^a^	852 ^b^	5 ^b^	2724 ^a^	864 ^b^	6 ^c^	6306 ^a^	1529 ^b^	1 ^c^	Balsamic, bitter almond	64,000	3,4
4-methyl-1-pentanol	1320	ND	ND	ND	ND	ND	ND	6 ^a^	2 ^b^	ND	Almond	5,000	4
1-Hexanol	1358	124 ^a^	46 ^b^	1 ^b^	130 ^a^	54 ^b^	1 ^c^	346 ^a^	134 ^b^	ND	Floral, green	110	1,6
3-Ethoxy-1-propanol	1363	5 ^a^	4 ^a^	ND	ND	ND	ND	ND	ND	ND	Fruity	100	2
*cis*-3-Hexen-1-ol	1369	5 ^a^	1 ^b^	ND	5 ^a^	2 ^b^	1 ^c^	ND	ND	ND	Fruity, green	400	6,2
1-Heptanol	1460	3 ^a^	2 ^b^	1 ^c^	3 ^a^	2 ^b^	1 ^c^	17 ^a^	10 ^b^	1 ^c^	Oily	2500	3
2-Ethyl-1-hexanol	1493	ND	ND	ND	3 ^b^	5 ^a^	ND	17 ^a^	18 ^a^	ND	Fruity, floral	8000	1,2
2,3-Butanediol	1547	9 ^a^	4 ^b^	2 ^c^	14 ^a^	3 ^b^	3 ^b^	5 ^a^	2 ^b^	2 ^b^	Fruity	150,000	2
1-Octanol	1561	13 ^a^	11 ^ab^	1 ^b^	7 ^a^	4 ^b^	1 ^c^	32 ^a^	13 ^b^	1 ^c^	Floral, fatty	40	1,3
1-nonanol	1639	5 ^a^	2 ^b^	1 ^b^	4 ^a^	4 ^a^	1 ^b^	43 ^a^	9 ^b^	1 ^c^	Fruity	600	2,7
1-Decanol	1765	14 ^a^	3 ^b^	2 ^b^	18 ^a^	3 ^b^	ND	24 ^a^	3 ^b^	ND	Fatty	400	3
1-Undecanol	1869	ND	ND	ND	3 ^a^	1 ^b^	ND	ND	ND	ND	Fruity	/	2
Benzyl alcohol	1883	11 ^a^	5 ^b^	ND	21 ^a^	16 ^b^	11 ^c^	53 ^a^	34 ^b^	12 ^c^	Fruity, floral	200,000	1,2
2-Phenylethanol	1918	930 ^a^	274 ^b^	185 ^b^	904 ^a^	672 ^b^	554 ^c^	2785 ^a^	688 ^b^	564 ^c^	Floral	10,000	1
Dodecanol	1970	10 ^a^	5 ^b^	1 ^c^	9 ^a^	2 ^b^	1 ^c^	ND	ND	ND	Fatty	7	3
Total higher alcohols		3688	1212	199	3848	1634	580	9705	2460	582			
Acids													
Acetic acid	1453	47 ^a^	4 ^b^	13 ^b^	21 ^a^	2 ^b^	ND	58 ^a^	4 ^c^	23 ^b^	Sour, vinegar	200,000	3,4
Isobutyric acid	1568	4 ^a^	ND	1 ^b^	2 ^a^	ND	2 ^b^	7 ^a^	2 ^b^	1 ^b^	Cheesy	200,000	3
Butanoic acid	1627	10 ^a^	ND	5 ^b^	8 ^a^	3 ^c^	5 ^b^	11 ^a^	2 ^c^	4 ^b^	Rancid, sweat	173	3
2-Methylbutanoic acid	1670	19 ^a^	ND	ND	8 ^a^	4 ^b^	ND	ND	ND	ND	Fatty, rancid	250	3
2-methylhexanoic acid	1671	ND	ND	ND	ND	ND	ND	36 ^a^	8 ^b^	8 ^b^	Rancid	/	3
Hexanoic acid	1845	542 ^a^	115 ^ab^	103 ^b^	414 ^a^	107 ^b^	91 ^c^	388 ^a^	81 ^b^	40 ^c^	Cheesy, fatty	420	3
Octanoic acid	2056	1440 ^a^	443 ^b^	189 ^b^	3094 ^a^	114 ^c^	289 ^b^	727 ^a^	216 ^b^	34 ^c^	Rancid, fatty	500	3
Decanoic acid	2275	145 ^a^	33 ^b^	17 ^b^	112 ^a^	101 ^b^	17 ^c^	175 ^a^	28 ^b^	15 ^c^	Rancid, fatty	1400	3
Total acids		2207	595	328	3659	331	404	1402	341	125			
Terpenics and C_13_-Norisoprenoids													
Linalool	1550	14 ^a^	8 ^b^	1 ^c^	9 ^a^	7 ^a^	1 ^b^	ND	ND	ND	Floral	25	1
α-Terpineol	1705	3 ^a^	2 ^b^	1 ^c^	4 ^a^	1 ^b^	ND	ND	ND	ND	Floral	250	1
Citronellol	1770	9 ^a^	5 ^ab^	ND	16 ^a^	8 ^b^	1 ^c^	17 ^a^	8 ^b^	1 ^c^	Floral	100	1
Nerol	1806	ND	ND	ND	3 ^a^	1 ^b^	ND	10 ^a^	ND	ND	Floral, citrus	400	1
β-damascenone	1831	37 ^a^	21 ^a b^	2 ^b^	14 ^a^	9 ^b^	1 ^c^	15 ^a^	6 ^b^	1 ^c^	Floral, fruity	0.05	1,2
Geraniol	1852	25 ^a^	5 ^b^	1 ^c^	12 ^a^	5 ^b^	1 ^c^	ND	ND	ND	Floral	20	1
Geranyl acetone	1860	10 ^a^	7 ^a^	1 ^b^	5 ^a^	2 ^b^	1 ^c^	5 ^a^	ND	ND	Floral	60	1
Nerolidol	2039	14 ^a^	3 ^b^	1 ^b^	ND	ND	ND	ND	ND	ND	Floral, fruity	100	1,2
Total		112	51	7	63	33	5	47	14	2			
Other compounds													
Benzaldehyde	1528	60 ^a^	13 ^b^	6 ^b^	110 ^a^	15 ^b^	1 ^c^	12 ^a^	4 ^b^	1 ^c^	Fruity	350	2
Methionol	1724	4 ^a^	1 ^b^	ND	2 ^a^	ND	ND	8 ^a^	1 ^b^	ND	Cooked potato, garlic	1500	6
Total		64	14	6	112	15	1	20	5	1			

Data are means ± SD (*n* = 3). Different letters represent significant differences at a significant level of 0.05. RI; calculated Retention Index. “/” means not found. “ND” means that the aroma compound is not detected or is found in trace amounts. RO; reverse osmosis technique, VD; vacuum distillation technique. ^a^ Odor descriptions are mainly obtained from the following literature: flavornet database (http://www.flavornet.org (accessed on 15 August 2021)) [9,57,58,59,60,61,62,63,64,65,66,67,68,69]. ^b^ Thresholds are mainly gotten from the following literature: [9,57,58,59,60,61,62,63,64,65,66,68]. ^c^ Each compound was attributed to 1 or more aroma class of sensory descriptors as follows: 1, floral; 2, fruity; 3, fatty; 4, pungent; 5, spicy; 6, vegetative; 7, earthy.

## Data Availability

Data sharing is not applicable to this article.

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
