# Peer review of "Comparison between Membrane and Thermal Dealcoholization Methods: Their Impact on the Chemical Parameters, Volatile Composition, and Sensory Characteristics of Wines"

_membranes, 2021, doi:10.3390/membranes11120957_

Round 1
Reviewer 1 Report
In recent years there has been an increasing trend of a part of consumers who want wines with a lower alcoholic strength. In particular, social demands related to road safety, and some health reasons have considerably increased the demand for dealcoholized wines (Margallo et al., 2015). The problem is that the dealcoholization of wines has two major drawbacks. The first is technical, since it is extremely difficult to dealcoholize a wine while maintaining its original organoleptic characteristics. Generally, both its aromatic qualities and its taste tend to worsen the greater the decrease in its alcoholic strength. The second drawback is legal, since the laws vary according to each country. The International Wine Organization (OIV) establishes that a percentage of alcohol greater than 2% v / v cannot be reduced. Partial dealcoholization (up to 2% v / v) has become another practice in oenology (Ferrarini et al., 2015).
Dealcoholization methods generate changes in the organoleptic properties of wines; even membrane methods cause compositional changes that go beyond alcohol extraction. This makes the resulting wines have a different flavor and aroma. However, scientific publications cite experiences in which excellent results have been achieved with dealcoholization of up to 2% v / v.
In the article, the physicochemical parameters, the composition of the volatile components and the sensory characteristics of a white, a rosé and a red wine dealcoholized by two methods, reverse osmosis and vacuum distillation, were compared. The final objective of the work is to determine the most suitable method for the dealcoholization of wines.
There are numerous references to data obtained from wines subjected to dealcoholization, as shown in the existing bibliography. Apart from article works there are numerous patents on this subject.
The results of this study are specific to wines produced from the Chardonnay, Pinot Noir and Merlot grape varieties that have been considered the best grapes in terms of sensory quality and aroma. It is not understood, because the wines have been subjected to a 0.7% v / v dealcoholization, when the regulations adjust to values ​​of 2% v / v and 5% v / v at least.
Section 4, Limitations, is the one that shows the reality of the research: Few samples of grapes (a single sample), Few varieties of grapes (three varieties of grapes), need for further trials, Incorporation into complementary evaluations, etc.
The conclusions reached are very basic and elementary since dealcoholization has transformed the original product into very different products and there are a significant number of references that indicate that the RO process is the most widespread and used.
Both RO and VD removed ethanol to a controllable degree, but also significantly affected physicochemical parameters. The results obtained in the article are not very relevant since other authors have already carried out this type of dealcoholization.
The original results obtained in this research are scarce due to the little news about comparing dealcoholization between two already mature techniques. It is not enough with these tests developed in this work, to draw conclusions related to the difficulties in the microbiological stabilization and the storage of dealcoholized wines.
The conclusions of this work show the level of research, which is fundamentally based on the carrying out of trials and tests of dealcoholized wines. The tests are very exhaustive and show the results of the physicochemical parameters, the composition of the volatile components and the sensory characteristics.
The companies in the sector bet on the RO method because it does not modify the characteristics of the wine to a lesser extent to extract the alcohol.
Volumetric losses, costs and complexity of processes, investment costs, etc. are not discussed at work.
Author Response
Response to Reviewer 1 Comments
Thank you so much for your intensive review of our manuscript. We have read all comments carefully and have made corrections which we hope would meet your approval. The corrections were made in our manuscript using the “Track Changes” function in Microsoft word. All comments and revisions are also shown below where responses to comments are marked in red.
Point 1: In recent years there has been an increasing trend of a part of consumers who want wines with a lower alcoholic strength. In particular, social demands related to road safety, and some health reasons have considerably increased the demand for dealcoholized wines (Margallo et al., 2015). The problem is that the dealcoholization of wines has two major drawbacks. The first is technical, since it is extremely difficult to dealcoholize a wine while maintaining its original organoleptic characteristics. Generally, both its aromatic qualities and its taste tend to worsen the greater the decrease in its alcoholic strength. The second drawback is legal, since the laws vary according to each country. The International Wine Organization (OIV) establishes that a percentage of alcohol greater than 2% v / v cannot be reduced. Partial dealcoholization (up to 2% v / v) has become another practice in oenology (Ferrarini et al., 2015).
Dealcoholization methods generate changes in the organoleptic properties of wines; even membrane methods cause compositional changes that go beyond alcohol extraction. This makes the resulting wines have a different flavor and aroma. However, scientific publications cite experiences in which excellent results have been achieved with dealcoholization of up to 2% v / v.
In the article, the physicochemical parameters, the composition of the volatile components and the sensory characteristics of a white, a rosé and a red wine dealcoholized by two methods, reverse osmosis and vacuum distillation, were compared. The final objective of the work is to determine the most suitable method for the dealcoholization of wines.
There are numerous references to data obtained from wines subjected to dealcoholization, as shown in the existing bibliography. Apart from article works there are numerous patents on this subject.
Response 1: Thank you for the suggestion. Please this has been incorporated into the introduction.
Point 2: The results of this study are specific to wines produced from the Chardonnay, Pinot Noir and Merlot grape varieties that have been considered the best grapes in terms of sensory quality and aroma. It is not understood, because the wines have been subjected to a 0.7% v / v dealcoholization, when the regulations adjust to values ​​of 2% v / v and 5% v / v at least.
Response 2: The statement “The results of this study are specific to wines produced from the Chardonnay, Pinot Noir and Merlot grape varieties that have been considered the best grapes in terms of sensory quality and aroma” is based on a general assumption (due to existing bibliography) that these varieties have good aroma and sensory quality, but not based on the results of this study. It has been re-written for better understanding.
Point 3: Section 4, Limitations, is the one that shows the reality of the research: Few samples of grapes (a single sample), Few varieties of grapes (three varieties of grapes), need for further trials, Incorporation into complementary evaluations, etc.
Response 3: You are right. We stated in our limitations that “It is possible that the dealcoholization of wine produced from terpene varieties such as Muscats, Gewürztraminer, Riesling, and Torrontés would have a greater positive influence on the overall sensory profile of those wines as these varieties have high contents of fruity and floral varietal aroma”. We have made it more clear in the limitation section.
Point 4: The conclusions reached are very basic and elementary since dealcoholization has transformed the original product into very different products and there are a significant number of references that indicate that the RO process is the most widespread and used.
Response 4: Yes, there are a significant number of references that indicate that the RO process is the most widespread and used, however most of these studies were not comparative studies. It is highly possible that in a comparative study, RO would not be the better one and such is one study which proved VD to be better than RO (Motta et al., 2017).
Point 5: Both RO and VD removed ethanol to a controllable degree, but also significantly affected physicochemical parameters. The results obtained in the article are not very relevant since other authors have already carried out this type of dealcoholization.
Response 5: Please we stand to be corrected on this. Only one study similar to ours (comparison) has been conducted (Motta et al., 2017) and the dissimilarity between the two studies is mainly on the level of dealcoholization, the type of wines used, and the kind of analyses conducted (i.e. Motta et al. (2017) never compared the two methods in terms of sensory characteristics as we did) which shows the novelty of our research. Moreover, with regards to individual studies using only one of these methods (either RO or VD), there are cases where dealcoholization at 0.5% v/v ethanol is reported (Petrozziello et al., 2019), however, no data on sensory was reported. Our study is the second study to the best of our knowledge that has compared these two methods in a single study and the only study that has also compared these two methods of dealcoholization (at 0.7% v/v ethanol) based on sensory characteristics. Where Motta et al. (2017) found vacuum distillation to be better than reverse osmosis, this study on the other hand, found RO to be better even at 0.7% v/v dealcoholization. Therefore, we strongly disagree that “The results obtained in the article are not very relevant”.
Point 6: The original results obtained in this research are scarce due to the little news about comparing dealcoholization between two already mature techniques. It is not enough with these tests developed in this work, to draw conclusions related to the difficulties in the microbiological stabilization and the storage of dealcoholized wines.
Response 6: Please the sentence has been modified as “From a practical perspective, wine producers using RO or VD in the dealcoholization of wine may need to consider the risks associated with lower concentration or loss of free sulfur dioxide (which can act as a preservative) or oxygen uptake (which can lead to oxidation during aging) during dealcoholization.
Point 7: Volumetric losses, costs and complexity of processes, investment costs, etc. are not discussed at work.
Point 7: Volumetric losses, costs, complexity of processes, investment costs are not the aims of the research, hence are not discussed.
Reviewer 2 Report
Authors present results of their work on comparing membrane and thermal dealcoholization methods. English needs serious revision e.g. in abstract “VD resulted in higher color intensity and visibly affected the color of dealcoholized 25 rosé and red wines…”. Or what does it mean in lines 32-33: decreased from 35% to 61%? Please explain. So in intro… already first sentence contains “content” word repeated 3 times – please rephrase and recheck English in the entire manuscript.
In other aspects intro is interesting and doesn’t require any changes.
In 2.1 add CAS numbers to all chemicals used and their purity grades.
Line 137: this is truly brief sentence - “Briefly, 5 L of 137 wine were dealcoholized.”.
Line 145: what was app. the flow rate value?
Description of methodologies – perfect work!
Results and their discussion is also comprehensive. I didn’t find any major or even minor mistakes in presentation of results and their description. It is good to see comprehensive discussion of results when journal doesn’t limit pages numbers so that authors can give their full opinion. I love figure 2.
Author Response
Response to Reviewer 2 Comments
Thank you so much for your intensive review of our manuscript. We have read all comments carefully and have made corrections which we hope would meet your approval. The corrections were made in our manuscript using the “Track Changes” function in Microsoft word. All comments and revisions are also shown below where responses to comments are marked in red.
Point 1: Authors present results of their work on comparing membrane and thermal dealcoholization methods. English needs serious revision e.g. in abstract “VD resulted in higher color intensity and visibly affected the color of dealcoholized 25 rosé and red wines…”. Or what does it mean
Response 1: “VD resulted in higher color intensity and visibly affected the color of dealcoholized rosé and red wines…” implies that the increase in color intensity was perceptible to the human eye as the total color difference (∆E*) in the rose and red wines was ≥ 3.0 (in Table 1). We have re-written the sentence for better understanding.
Point 2: lines 32-33: decreased from 35% to 61%? Please explain.
Response 2: Please the sentence has been re-written for better understanding.
Point 3: So in intro… already first sentence contains “content” word repeated 3 times – please rephrase and recheck English in the entire manuscript.
Response 3: Please changes have been made and the English used in writing the manuscript rechecked.
Point 4: In other aspects intro is interesting and doesn’t require any changes.
Response 4: Thank you. We are humbled.
Point 5: In 2.1 add CAS numbers to all chemicals used and their purity grades.
Response 5: Please CAS numbers to all chemicals used and their purity grades have been mentioned in section 2.1.
Point 6: Line 137: this is truly brief sentence - “Briefly, 5 L of 137 wine were dealcoholized.”.
Response 6: Please the sentence has been re-written.
Point 7: Line 145: what was app. the flow rate value?
Response 7: The flow rate was 70 mL/min. Please this has been mentioned in the manuscript.
Point 8: Description of methodologies – perfect work!
Response 8: Thank you.
Point 9: Results and their discussion is also comprehensive. I didn’t find any major or even minor mistakes in presentation of results and their description. It is good to see comprehensive discussion of results when journal doesn’t limit pages numbers so that authors can give their full opinion. I love figure 2.
Response 9: Thank you.
Round 2
Reviewer 1 Report
There are no new comments